# Cognitive functioning in context: Leisure activity engagement, social capital, and urbanicity-rurality interplay

B. Paige Trubenstein[1,2]*, Shandell Pahlen[1,3], Robin P. Corley[3], Sergio Rey[4,5], Sally J. Wadsworth[3], Chandra A. Reynolds[1,3,6]

1 University of California, Riverside – (United States of America), Department of Psychology, Riverside, California, United States of America, 2 Angelo State University – (United States of America), Department of Psychology, San Angelo, Texas, United States of America, 3 University of Colorado, Boulder – (United States of America) - Institute for Behavioral Genetics, Boulder, Colorado, United States of America, 4 University of California, Riverside – (United States of America) – School of Public Policy, 900 University Ave, Riverside, California , United States of America, 5 San Diego State University – (United States of America) - Department of Geography, Campanile Drive, San Diego, California, United States of America, 6 University of Colorado, Boulder – (United States of America) - Department of Psychology and Neuroscience, Boulder, Colorado, United States of America

* paige.trubenstein@angelo.edu

## Abstract

Leisure activity associations with cognition may operate differently depending on an individual's context. We evaluated whether activity-cognition associations were influenced by community and geographic features in CATSLife (*M*age = 33.17 years, N = 1201). Measures included cognition indexed by IQ and activity engagement indexed by time, cognitive demand, and frequency. County-level Index of Relative Rurality (IRR) and Social Capital Index (SCI), i.e., the availability of social networks and resources, captured environmental features. In multilevel models, activity engagement was more strongly associated with IQ than SCI and rurality. We found evidence that activity-IQ associations were magnified in urban environments when SCI was high, but associations were reduced when SCI was low. However, adolescent IQ diminished associations, revealing selection effects. Our findings highlight interrelated individual, community, and geographic factors influencing cognitive functioning, but also the saliency of earlier life cognition to attained contexts, that together may contribute to cognitive maintenance at midlife and beyond.

## Introduction

In a world with aging populations [1] coupled with the concern of aging well (i.e., free of neurocognitive dysfunction), elucidating the role of the person and place influences in cognitive functioning is critical. Much research has been devoted to exploring the cognitive differences that exist based on individual activity engagement [2] and residence [3–5]. Both perspectives have important and practical implications for public

**Data availability statement:** The minimal data set underlying the findings reported in this manuscript, including values behind reported means, standard deviations, and other measures, as well as data used to generate figures, is available by request for access, and will require completion of a data use agreement, documentation of training in the protection of human subjects, and a statement of intended use. Such requests can be directed to Dr. Daniel Gustavson (Institute for Behavioral Genetics, University of Colorado Boulder; Email: Daniel. gustavson@colorado.edu). The data contain potentially identifying or sensitive information, and sharing may be restricted according to participant consent and IRB directives. Analysis scripts and documentation of data coding procedures are provided in the Open Science Framework repository at: https://osf.io/xuetg (DOI: https://doi.org/10.17605/OSF.IO/XUETG).

**Funding:** The authors gratefully acknowledge support from the National Institutes of Health, NIH AG046938 [MPIs, to C.R. and S.W.]. The funders had no role in study design, data collection and analysis, decision to publish, or preparation of the manuscript.

**Competing interests:** The authors have declared that no competing interests exist.

health, but health recommendations typically divorce the individual from the environment which undermines the realities of person *and* place effects. For example, the 2020 Lancet report on reducing the dementia burden centers most interventions around the individual such as increasing cognitive engagement [6]. Urban places and community features in which an individual lives and navigates have been shown to be correlated with cognitive functioning [7]. However, environments are not randomly distributed, and individuals can have an active role in the selection of communities that benefits their own characteristics [8,9]. We aim to disentangle how the individual embedded in a larger environment may impact cognitive functioning by examining the interplay between individual leisure activity engagement, community, and the urban-rural continuum.

A core tenant of the Bioecological model is that development is not merely the result of static traits or isolated environmental factors but rather emerges from mutual interactions between individuals and their environments across different systemic levels [10]. Activity engagement reflects one level of the dynamic interplay between an individual and her/his environment [10] and is theorized to increase cognitive functioning and dampen the rate of later cognitive decline through processes that promote compensatory strategies [11,12] or contribute to cognitive reserve [13]. Cognitive reserve is conceptualized as the brain's capacity to flexibly recruit and optimize cognitive processes, accounting for individual differences in vulnerability to cognitive impairment and is often measured through proxies such as IQ, educational attainment, occupational complexity, and even leisure and physical activity [13]. Activity engagement is associated with better cognitive functioning and maintenance across time, with stronger evidence existing for physical activities than social or cognitive activities [2,14]. Although the direct effect of any type of individual activity engagement on cognition is expected to be small, lifestyle and environmental factors may accumulate to influence cognitive health [2,15]. One such environmental factor that reflects the collective resources and engagement within the community is social capital [3]. Social capital represents the social networks available to an individual as well as tangible and emotional resources, trust, and interchange opportunities that are developed through those social networks [16]. A recent paper that coined the term 'cognability' of neighborhoods, or structural features surrounding an individual that either promote cognitive maintenance or accelerate cognitive aging [7], incorporated many community features that overlap with social capital resources [17]. The authors found that access to recreation centers and museums (but not parks or libraries), and civic and social organizations, were moderately to strongly predictive of cognitive functioning in older adults residing in metropolitan areas [7]. However, there is yet no clear application of 'cognability' to either capture the activity engagement of individuals or the community features in rural regions.

Literature evaluating urban-rural activity engagement differences is growing with gaps emerging based on the type of activity. Across rural and urban individuals research suggests that rural individuals report significantly less physical activity than their urban counter parts [18–20]. A smaller literature on social activity engagement suggests that rural individuals engage in significantly fewer social

activities than urban individuals [21,22]. Urban environments may provide more cognitive stimulation [4,23,24]; however, little research has examined geographic differences in cognitive activity engagement nor how they may interact with social capital. Engagement in cognitive and social activities may stem from childhood or young adult cognitive ability and educational attainment differences [2,25,26] suggesting that activity engagements are the outcomes of early life factors and abilities. Hence, any advantages to cognitive functioning from cognitive and social activity engagement may arise due to self-selection into the activities. However, clearer direct benefits of physical activity, beyond selection factors, have been reported in some studies [2,14]. As activity engagement associations with geographic contexts are often not assessed, the presence of selection effects remains unclear, but are essential to assess inasmuch as some studies report a facilitating effect of geographic contexts for activity and cognitive functioning (e.g., social activity and neighborhood cohesion) [27].

Studies have suggested that rural areas have higher amounts of social capital [28–32], whereas others note geographic differences depending on the aspect of social capital considered. For example, individuals in rural areas have been found to have higher levels of social trust [28,30] although this is debated [33], stronger family ties [34], participation in the local community [30,33], and neighborhood connections [30]. Alternatively, individuals in urban areas report higher levels of social agency, such as being proactive at work or seeking mediation for disputes with neighbors [30] and report greater tolerance of diversity such as valuing multiculturalism [30]. Much of the prior literature has evaluated these rural-urban differences in social capital using a categorical/typological representation of rurality [20,28,32,34,35] rather than considering it on a continuum that provides added information otherwise obscured with typologies [36,37].

Our study aims to extend past work to understand the urban-rural health divide by examining how activity engagement, social capital, and rurality may contribute or function together to influence cognitive function past young adulthood. Ages spanning 30 up to 45 years, also known as established adulthood [38], is a time ripe for cognitive health interventions prior to the midlife transition when early cognitive changes may begin to show [39,40]. Whereas 'cognability' of neighborhoods focuses on older adults, we consider individuals approaching midlife from the first completed assessment of the Colorado Adoption/Twin Study of Lifespan behavioral development and cognitive aging (CATSLife) [41,42]. CATSLife provides a valuable opportunity to incorporate rich cognitive and behavioral assessments and continuous geospatial measures of social capital [17,43] and geography as represented by Index of Relative Rurality (IRR) [44]. Our study will explore individual activity engagement and geospatial influences on cognitive functioning, as indexed by IQ via four research questions: 1) Is IQ associated with geospatial residential features indexing social capital or rurality? 2.) Do geospatial-IQ associations diminish post-SES adjustment, indicative of neighborhood selection? 3) What is the activity-IQ relationship after considering social capital or rurality? 4) Does the level of rurality moderate activity-IQ or social capital and IQ associations? Within these aims, we will examine the extent to which earlier life cognitive differences contribute to activity, geospatial, and geographical associations with cognitive function, revealing potential individual selection.

## Method

This research was conducted with approvals provided by the respective Institutional Review Boards at the University of Colorado, Boulder [14–0421] and the University of California, Riverside, [HS 14–073] and was conducted according to the APA ethical standards.

## Participants

The current study uses data from the Colorado Adoption/Twin Study of Lifespan behavioral development and cognitive aging (CATSLife) [41,42], with participant recruitment between July 1, 2015 and March 31, 2021 (N = 1,327, $M$age = 33.45 years, $SD$ = 5.04). Participants provided written informed consent. Over 96% of the sample participated before the COVID-19 pandemic. CATSLife includes twin and adoptee/non-adoptee participants from two longitudinal developmental studies, the Longitudinal Twin Study (LTS) [45] and the Colorado Adoption Project (CAP) [46,47].

The majority of CATSLife participants (N = 1,221 or 92%) had IQ scores. The analytic sample is comprised of United States residents with available geographic measurements, sociodemographic measures, and IQ scores for a total N = 1201 (F = 53%). On average, participants were 33.17 years old (SD = 4.96, range = 28–49). The majority of the sample identified as White (92%), and non-Hispanic (94%), representative of Colorado state demographics at the time of initial cohort recruitment (see Table 1).

## Measures

### Cognitive ability

Current cognitive ability was assessed using the WAIS-III via three measures: Full-scale (FSIQ), Verbal (VIQ), and Performance (PIQ) IQ scaled scores [48]. Reliability has been found to be excellent for FSIQ and the other two scaled scores (0.88 to 0.98) [48]. For reporting, we prioritized FSIQ.
*Adolescent IQ.* Adolescent general cognitive ability comes from archival year 16 assessments given to the CAP and LTS samples. For both samples, adolescent IQ (IQ16; N = 1,152) was assessed approximately at 16.59 (SD = 4.99) years and indexed by the WAIS full-scale (FSIQ), verbal (VIQ), and performance scaled scores (PIQ) using the WAIS-R [49] or WAIS-III [50].

### Cognitive engagement

**Hours Per Week (HPW).** Participants self-reported weekly time spent on specific activities from an adapted 20-item engagement questionnaire for adults [51]. Hours per week (HPW) were rescaled: "None" = 0, "less than 1 hour" = 1, "2-3 hours" = 2.5, "4-5 hours" = 4.5, "6-7 hours" = 6.5, "8 or more hours" = 8. See S1 Table in S1 File for item-level descriptives. Three cognitive activities were identified based on a prior theorical and empirical work [52,53] and summed together for

**Table 1. Descriptive statistics of key variables.**

| | N | Mean | SD | Var. | Range | Min | Max |
|---|---|---|---|---|---|---|---|
| Age | 1201 | 33.17 | 4.96 | 24.64 | 21.28 | 28.05 | 49.33 |
| Female | 1201 | 0.53 | 0.5 | 0.25 | 1 | 0 | 1 |
| White | 1201 | 0.92 | 0.27 | 0.07 | 1 | 0 | 1 |
| Hispanic | 1201 | 0.06 | 0.23 | 0.05 | 1 | 0 | 1 |
| Educational attainment | 1191 | 16.84 | 2.94 | 8.65 | 11 | 11 | 22 |
| Occupational attainment | 1177 | 5.96 | 1.54 | 2.36 | 6 | 2 | 8 |
| FSIQ | 1201 | 110.41 | 11.88 | 141.08 | 79 | 69 | 148 |
| VIQ | 1201 | 107.42 | 11.51 | 132.50 | 72 | 70 | 142 |
| PIQ | 1201 | 112.68 | 13.54 | 183.31 | 87 | 68 | 155 |
| HPW Cognitive | 1189 | 4.68 | 3.79 | 14.37 | 24 | 0 | 24 |
| HPW Cognitive (LN) | 1189 | 1.50 | 0.72 | 0.52 | 3.22 | 0 | 3.22 |
| Average Cognitive Demand of Hobbies | 741 | 2.56 | 0.63 | 0.4 | 3.41 | 1.59 | 5 |
| Average Cognitive Demand of Hobbies (LN, centered) | 741 | 0.22 | 0.23 | 0.05 | 1.15 | −0.23 | 0.92 |
| Number of Hobbies | 741 | 2.21 | 1.47 | 2.17 | 11 | 1 | 12 |
| Number of Hobbies (LN, centered) | 741 | 0.08 | 0.39 | 0.15 | 1.87 | −0.31 | 1.56 |
| Social Capital Index | 1201 | −0.48 | 0.65 | 0.43 | 5.76 | −2.42 | 3.34 |
| Index of Relative Rurality | 1201 | 0.35 | 0.11 | 0.01 | 0.64 | 0.04 | 0.68 |

*Note:* N reflects analytic sample. FSIQ = Full-Scale IQ, VIQ = Verbal IQ, PIQ = Performance IQ, HPW = Hours per week. LN = Natural log-transformed. The N for Average Cognitive demand and Number of Hobbies reflects participants who reported any time engaging in hobbies per week.

total HPW: "reading for fun?", "spending time on a hobby?", and "playing a musical instrument". Correlations supported positive, small associations amongst the three items ($r$ range = 0.14 to 0.23). On average, people reported spending approximately 4.7 hours per week ($SD$ = 3.8) on cognitive engagement (see Table 1). Total HPW scores were natural log-transformed to adjust for skew.

**Cognitive demand.** Participants who reported spending any time on a hobby from the activity questionnaire item (N = 871) then self-reported their specific hobbies (N = 741). Individuals missing on self-reported hobbies (N = 130; 15%) did not significantly differ on sociodemographics (all $p \geq 0.39$), but had significantly lower educational attainment, occupational attainment, and IQ scores (see S2 Table in S1 File).

Each hobby was rated on a 5-point cognitive demand scale informed by prior work [54–56]: 1 = absolutely no demand [e.g., sleeping], 2 = some, 3 = moderate, 4 = a lot, and 5 = high [e.g., writing music]. Activity ratings were blind dual coded by research assistants and final ratings were selected after consensus review that included 5 members, including authors BPT, SP, and CAR. Hobby demand coding was somewhat consistent with 62.0% of raters in absolute agreement and 95.4% within 1 point for cognitive rating scores. Cognitive demand scores were averaged across all hobbies using the geometric mean to minimize outlier influence [57]. To adjust for skew, demand scores were natural log-transformed and centered on 0.693 or a log-transformed equivalent demand score of 2; descriptives are reported in Table 1.

**Number of hobbies.** Participants could list any number of their hobbies with the maximum total reported of 12 ($M$ = 2.2, $SD$ = 1.5, Skew = 1.8). Number of hobbies was natural log-transformed to account for skew, centered on 1.

## Community and environmental measures

Participant-provided address information was geocoded to county level Federal Information Processing Standards (FIPS) identifiers based on 2010 US Census Geocoder and supplemented by online mapping tools (Google Maps) for a few missing cases. Altogether, 59.23% of the sample population lived in Colorado with 47 states represented; the number of individuals in a given county-state ranged from 1 to 105 individuals. FIPS codes were matched to publicly available databases on rurality and social capital described below.

**Social Capital Index (SCI).** We utilized the 2014 county-level social capital index (SCI) from the Northeast Regional Center for Rural Development [17,43]. The SCI is the principal component of several facets of social capital, adjusted for population size: 1) the numbers of business, political, civic and professional organizations, and other resources (e.g., bowling alleys, religious organizations, fitness centers, etc.); 2) community engagement (e.g., census and voter turnout), and 3) quantity of domestic non-profit organizations. Further details on the individual components involved in the construction and development of SCI can be seen in [17]. SCI was standardized to a t-score distribution ($M$ = 50, $SD$ = 10), according to the CATSLife sample, and centered on 50 for analyses.

**Index of Relative Rurality (IRR).** The publicly available IRR provides county-level relative rurality scores across the United States for 2000 and 2010 [44]. Comparative work evaluating different place-based metrics of rurality to IRR show moderate to strong correspondence to other rank-based and multidimensional measures of the urban-rural continuum ($r$ range = 0.59 to.089) [58]. Our analysis used the 2010 version of the IRR, closest in time to when CATSLife members participated. The IRR is an unweighted average of the logarithm transformed population size, the logarithm transformed population density, percent of urban population and remoteness (distance from the centroid of the county to nearest Metropolitan/Micropolitan Statistical Area) [37,58,59]. The IRR measures an area's relative rurality using basic dimensions of environmental qualities and scores the area on a bounded scale ranging from 0 (completely urban) to 1 (completely rural) [36,37,58,59]. IRR was mean centered at 0.35 (see Table 1).

**Live together.** We created the dummy coded live together covariate (1 = living together) to account for the few siblings who lived in the same residence (3.8%).

### Educational attainment

Participants self-reported their educational attainment on a 11-point scale ranging from "Less than high school diploma or GED" = 11 to "Advanced degree (e.g., doctorate, M.D., law degree)" = 22 [60]. Educational attainment was centered on 18 or a bachelor's degree.

### Occupational attainment

Participants self-reported their current or most recent occupational attainment via an adapted Hollingshead 8-point scale, ranging from homemakers (score = 1) to executive/professional with advanced degree (score = 8) [61]. Homemakers and students (N = 64) were recategorized as semi-skilled workers [61,62]. Those missing on occupational attainment (N = 50) were matched based on self-reported job titles. Occupation was centered on 6 or semi-professional.

### Covariates

Several covariates were included: age (centered at 33 years), project (LTS = 0, CAP = 1), sex (male = 0, female = 1), adopted status (non-adopted = 0, adopted = 1), ethnicity (0 = non-Hispanic, 1 = Hispanic), race (0 = non-White, 1 = White), live together (0 = living apart, 1 = living together), educational attainment (centered at 18 or BA/BS degree), and occupational attainment (centered at 6 or semi-professional).

### Statistical analyses

Multilevel models were fitted using PROC Mixed in SAS 9.4 (SAS Institute Inc., Cary, NC) to evaluate the associations of activity engagement, social capital, and rurality with cognitive functioning. Full-information maximum-likelihood was used to account for missingness. Differential sibling relatedness was accounted for both between ($\sigma^2$BW) and within ($\sigma^2$WI) sibships by estimating separate random effects for those siblings in adoptive, non-adoptive, or twin families (dizygotic, DZ; monozygotic, MZ).

We addressed environmental selection effects in two ways. First, by estimating sibling relatedness in activity engagement and geographic indices via intraclass correlations (ICC) by sibling type. In all tests, we observed similar ICC patterns after removing half-siblings, additional biological siblings in adoptive families, and siblings that live together (see S3 Table in S1 File). The ICC differences between sibling types allow us to infer genetic and environmental influences in the engagement and geographic measures. In other words, if MZ siblings' ICC is stronger than all other groups, this suggests some genetic influence. Alternatively, if ICCs are equivalent across sibling types this would suggest a stronger shared environmental effect. The second method was to check the robustness of activity engagement and geographic effects on IQ after accounting for similarity among siblings or socioeconomic (SES) indicators. Of note, we tested for similarity by geographic locale, clustering between and within the FIPS identifiers, recoded to state-level for those outside of Colorado given dispersion and retaining county-level FIPS within Colorado; but FIPS explained less than 1% of the variance in IQ outcomes compared to over 33% accounted by sibship across all family types.

We next examined the influence of individual engagement and geography on cognition. Models first included cognitive engagement and SCI on IQ. Our second model then added a broader geography of the urban-rural continuum as indexed by IRR. The final models assessed whether cognitive engagement or community varied by IRR. The same model strategy was followed for the three IQ measures, with FSIQ prioritized and reported in the main text. In addition, we repeated analyses for the cognitive engagement measures testing either a quantitative (HPW Cognitive) or qualitative (cognitive demand and number of unique activities) measure. Finally, to evaluate the robustness of the effect of activity, geography, and interaction of the two on cognitive function, we completed a series of sensitivity analyses accounting for earlier life cognitive differences. Sensitivity models were an extension of the best-fitting model in which we added IQ16 and the age difference between assessments, centered at 15 years. Model fit was assessed with standard fit indices of the chi-square difference test ($\Delta\chi2$) for nested models and the Akaike's Information Criterion (AIC) [63].

## Results

### Descriptive statistics

Descriptives for key variables are reported in Table 1. CATSLife participants tend to perform higher on average on FSIQ ($M=110.41$, $SD=11.9$), VIQ ($M=107.42$, $SD=11.5$), and PIQ ($M=112.68$, $SD=13.5$), than the general population ($M=100$, $SD=15$). Overall, participants tend to live in counties with lower social capital (SCI; $M=-0.47$) and there is less variability across the sample ($SD=0.66$) than the national average ($M=0.01$, $SD=1.26$) [43]. Participants also live in more urban than rural areas (IRR $M=0.35$, $SD=0.11$) than the national average ($M=0.50$, $SD=0.10$) [44]. Fig 1 illustrates the spread of SCI by IRR, with county references. As an example, Boulder County, Colorado (SCI = −0.05, IRR = 0.39) is a county close to the national average of SCI and IRR.

### Is IQ associated with geospatial residential features indexing social capital or rurality?

To evaluate associations under our first research question, we calculated partial correlations and ICCs by sibling type, after accounting for sociodemographics (see Table 2). Nearly all activity engagement measures were modestly correlated with each IQ measure ($rs=0.11$ to $0.27$, $ps<.002$). ICCs for HPW suggested possible nonadditive genetic effects as identical twins (MZ) were more than twice as similar as fraternal (DZ) and other full siblings. ICCs for cognitive demand and number of hobbies were not significantly different across sibling groups ($\Delta\chi^2(6)< 7.1$, $p>.31$). These patterns may arise from greater nonshared environmental influences or greater missingness in reporting among pairs for the cognitive demand subsample. In other words, the nonsignificant difference in the MZ ICC for cognitive demand, although comparable in size to HPW, could result from fewer numbers of intact sibling pairs with available qualitative data.

We found SCI was not correlated with any IQ measure. ICCs for SCI did not significantly differ between twins ($\Delta\chi^2(2)=2.3$, $p=.32$), however ICCs did differ between twin and control ($\Delta\chi^2(2)=18.7$, $p<.001$), and control and adopted ($\Delta\chi^2(2)=23.6$, $p<.001$) sibling groups. The ICC patterns suggest a twin environmental effect, or that twins' greater similarity

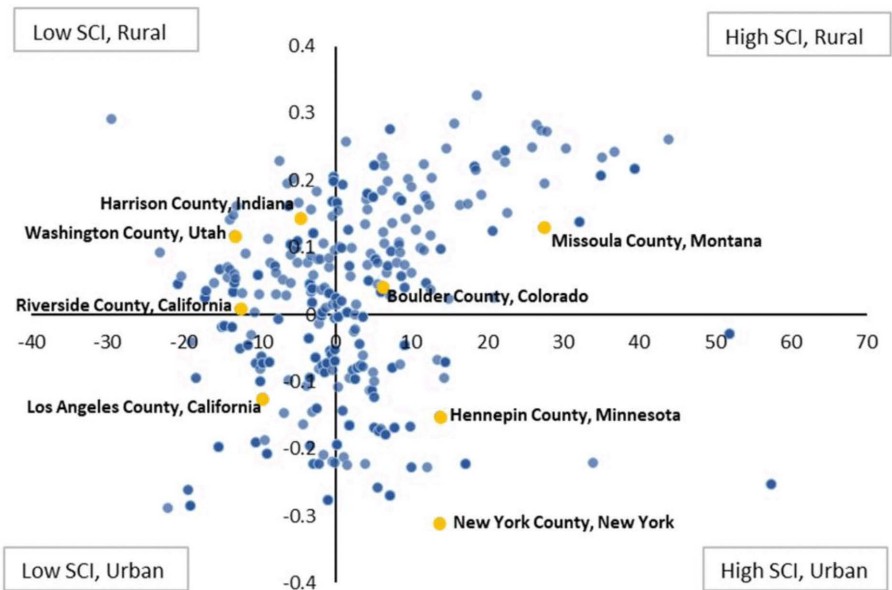

**Fig 1. Scatterplot of Social Capital (SCI) and Index of Relative Rurality (IRR). Legend:** SCI is plotted on the x-axis and IRR is plotted on the y-axis. Color saturation reflects greater representation. SCI is standardized to a t-score distribution ($M=50$, $SD=10$) and centered at 50 as shown; IRR is centered on 0.35.

**Table 2. Partial correlation and intraclass correlation coefficients (ICC).**

| | | FSIQ | VIQ | PIQ | ICCs | | | |
| --- | --- | --- | --- | --- | --- | --- | --- | --- |
| | | | | | MZ | DZ | Con | AD |
| **HPW** | | **.26** | **.27** | **.19** | **0.34** | 0.07 | 0.12 | 0.01 |
| N = 1189 | p | <.0001 | <.0001 | <.0001 | | | | |
| **Demand** | | **.14** | **.12** | **.11** | 0.33 | 0.12 | 0.04 | 0.00 |
| N = 741 | p | .0002 | .001 | .003 | | | | |
| **Number** | | **.14** | **.18** | .05 | 0.00 | 0.18 | 0.00 | 0.00 |
| N = 741 | p | .0002 | <.0001 | .192 | | | | |
| **SCI** | | .04 | .04 | .02 | **0.37** | **0.47** | 0.07 | **0.15** |
| N = 1201 | p | .140 | .122 | .416 | | | | |
| **IRR** | | **−.15** | **−.16** | **−.10** | 0.39 | 0.33 | 0.29 | 0.18 |
| N = 1201 | p | <.0001 | <.0001 | .001 | | | | |

*Note.* N's reflect those with IRR (Index of Relative Rurality) and IQ scores. Correlations adjusted for sex, age, project, adopted status, race, ethnicity, and live together. FSIQ = Full-Scale IQ; VIQ = Verbal IQ; PIQ = Performance IQ; ICC$^R$ = ICC excluding up to 25 half-siblings in control or additional biological siblings in adoptive families and up to 46 participants that live together; HPW = Log-transformed cognitive hours per week; Demand = Log-transformed average cognitive demand of reported hobbies; Number = Log-transformed number of cognitive hobbies; Sibling type: MZ = monozygotic twins; DZ = dizygotic twins; Con = individuals in nonadoptive ("control") families; AD = individuals in adoptive families. Bolded = $p < .05$.

on their SCI is not explained by genetic contributions but rather shared environmental factors related to being a twin. Importantly, the greater twin similarity was not attributable to SCI mean differences between sibling groups, $t(1199) < 0.77$, $p > .44$.

IRR was significantly correlated with all IQ measures ($rs = −0.10$ to $−0.16$, $p < .001$), indicating people living in more urban places tend have to higher IQ scores. ICCs between sibling groups for IRR did not significantly differ, $\Delta\chi^2(6) = 6.6$, $p = .36$, suggesting shared environmental experiences influencing rurality of an individual's residence.

### Do geospatial-IQ associations diminish post-SES adjustment, indicative of neighborhood selection?

To address our second research question, we examined potential socioeconomic status (SES) selection effects in activity engagement, SCI, and IRR associations with FSIQ; all models accounted for sociodemographics and clustering among siblings. Partial correlations among education, occupational complexity, cognitive engagement, social capital and rurality variables are reported in S4 Table in S1 File. Standardized parameters estimates before and after adjustment of educational and occupational attainment are reported in Fig 2, with full model parameters available in S5 and S6 Tables in S1 File. All measures were associated with FSIQ, except for SCI ($\beta = 0.03$, $p = .17$). Activity engagement associations remained consistent after SES adjustment. In contrast, IRR-FSIQ associations were no longer significant and were moderately attenuated by over 67% post-SES adjustment. Attenuation in the IRR association was more attributable to educational attainment ($B = 1.32$, $p < .001$) than occupation ($B = 0.37$, $p = .06$), see S5 Table in S1 File. Similar SES attenuation patterns were observed for VIQ and PIQ (see S7-S10 Tables in S1 File). In summary, with respect to our second research question, selection effects were observed whereby attenuation of IRR associations occurred with entry of individual level SES indices, and via patterns of sibling similarity for cognitive engagement, social capital, and rurality.

### What is the activity-IQ relationship after considering social capital or rurality?

To address our third research question, we evaluated associations of cognitive engagement adjusting for SCI and IRR and accounting for covariates. All cognitive engagement measures were significantly associated with FSIQ after accounting for SCI and IRR (See Table 3 Models 1 and 2, respectfully); parameter estimates for all IQ measures can be seen in S11-S16

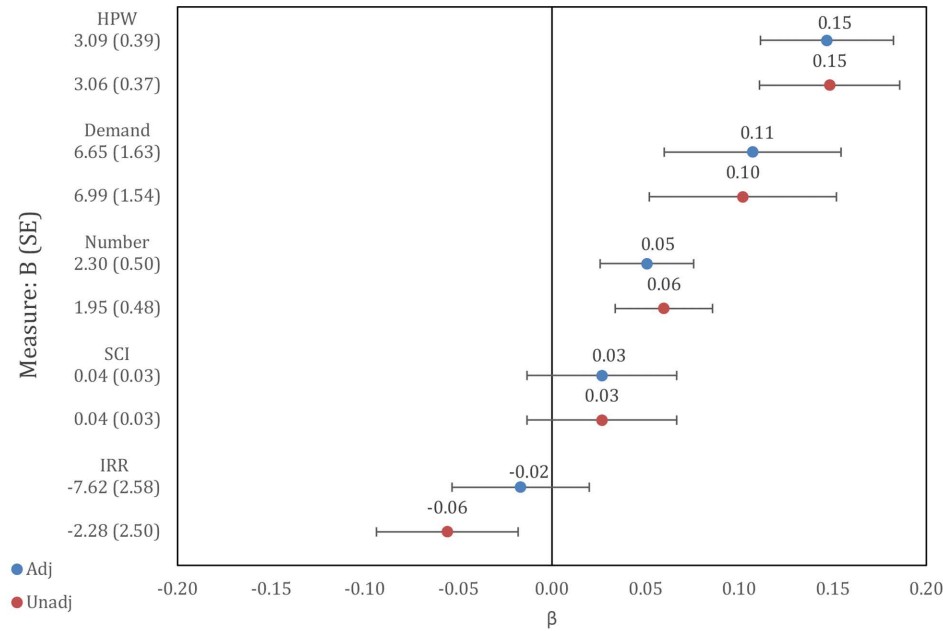

**Fig 2. Parameter estimates of key variables on FSIQ, pre- and post-SES adjustment. Legend:** Mixed-effects parameter estimates reported account for clustering among siblings, sex, age, project, adopted status, race, ethnicity, and live together. Standardized estimates are plotted in red and blue for parameters unadjusted/adjusted [Unadj/Adj] for educational and occupational attainment, respectively. On the y-axis under each measure, we report the unstandardized estimates with standard errors. FSIQ = Full-Scale IQ; HPW = Log-transformed cognitive hours per week; Demand = Log-transformed average cognitive demand of reported hobbies; Number = Log-transformed number of cognitive hobbies; SCI = Social Capital Index; IRR = Index of Relative Rurality.

Tables in S1 File. In these model tests, social capital and IRR were not significantly associated with FSIQ ($p >= 0.11$) or any other IQ measures after accounting for covariates and cognitive engagement ($p >= 0.10$; see Models 1 and 2 in S13-S16 Tables in S1 File for VIQ and PIQ). After controlling for SCI and IRR, we found 15 minutes of hobby engagement per day, or 1.75 total log-hours per week, was associated with a fifth of standard deviation higher FSIQ scores (Cohen's $d = .21$) compared with no time spent on engaging on any hobbies. Note that FSIQ effect sizes were calculated according to the population expected standard deviation of 15 and HPW is calculated as LN(HPW + 1). Similar patterns were seen for number hobbies and cognitive demand. For example, individuals that participated in 2 hobbies that were on average moderately demanding showed a comparable FSIQ difference (Cohen's $d = .28$) to those that participate in one hobby with some cognitive demand. We note that analyses with qualitative measures were restricted to those that reported their specific hobbies, thus minimum reports represent those with at least 1 hobby that was at least somewhat cognitively demanding. Engagement patterns were replicated with other IQ measures (see Supplemental Materials in S1 File), apart from PIQ where number of hobbies was not significant (B = 0.91, $p = .40$) (see S16 Table in S1 File).

In summary, with respect to our third research question, our results suggest that cognitive engagement measures (HPW, Demands of and Number of hobbies) association with FSIQ persist irrespective of locale, or that positive cognitive-activity associations are not dampened in places that differ in level of rurality or community resources. Likewise, activity engagement patterns were replicated for nearly all other tests with VIQ and PIQ measures.

### Does the level of rurality moderate activity-IQ or social capital and IQ associations?

To address our fourth research question, we evaluated interactions of IRR with cognitive engagement measures and of IRR with SCI.

**Table 3. Parameter estimates and model fit for FSIQ.**

| | Model 1 | | Model 2 | | Model 3 | | Model 4 | |
|---|---|---|---|---|---|---|---|---|
| | **B** | **se** | **B** | **se** | **B** | **se** | **B** | **se** |
| **HPW Cognitive** | | | | | | | | |
| Intercept | **107.78** | 1.59 | **107.73** | 1.59 | **107.73** | 1.59 | **107.55** | 1.59 |
| HPW | **3.03** | 0.37 | **3.07** | 0.37 | **3.07** | 0.37 | **3.15** | 0.37 |
| SCI | 0.03 | 0.03 | 0.03 | 0.03 | 0.03 | 0.03 | 0.04 | 0.03 |
| IRR | - | - | -3.72 | 2.47 | -2.76 | 5.84 | -2.11 | 2.54 |
| HPW* IRR | - | - | - | - | -0.61 | 3.39 | - | - |
| SCI* IRR | - | - | - | - | - | - | **-0.48** | 0.18 |
| Model Fit | | | | | | | | |
| -2 LL | 8568.5 | | 8566.3 | | 8566.3 | | 8559.5 | |
| AIC | 8608.5 | | 8608.3 | | 8610.3 | | 8603.5 | |
| Model Comparison | M1 | | M1-M2 | | M2-M3 | | M2-M4 | |
| $\Delta\chi^2$ (df) | | | 2.2 (1) | | 0 (1) | | 6.8 (1) | |
| p | | | .14 | | 1.0 | | .009 | |
| **Cognitive Demand** | | | | | | | | |
| Intercept | **110.43** | 2.03 | **110.32** | 2.03 | **110.27** | 2.03 | **110.05** | 2.02 |
| Demand | **7.51** | 1.54 | **7.37** | 1.54 | **7.28** | 1.54 | **7.13** | 1.53 |
| Number | **2.96** | 0.85 | **2.98** | 0.85 | **2.99** | 0.85 | **3.08** | 0.85 |
| SCI | 0.06 | 0.04 | 0.07 | 0.04 | **0.08** | 0.04 | **0.10** | 0.04 |
| IRR | - | - | -5.22 | 3.15 | **-9.34** | 4.42 | -3.80 | 3.17 |
| Cog Demand* IRR | - | - | - | - | 16.59 | 12.37 | - | - |
| SCI* IRR | - | - | - | - | - | - | **-0.69** | 0.26 |
| Model Fit | | | | | | | | |
| -2 LL | 5406.1 | | 5403.4 | | 5401.7 | | 5396.5 | |
| AIC | 5446.1 | | 5445.4 | | 5445.7 | | 5440.5 | |
| Model Comparison | M1 | | M1-M2 | | M2-M3 | | M2-M4 | |
| $\Delta\chi^2$ (df) | | | 2.7 (1) | | 1.7 (1) | | 6.9 (1) | |
| p | | | .100 | | .192 | | .009 | |

*Note.* FSIQ = Full-Scale IQ; HPW = Log-transformed cognitive hours per week; SCI = Social Capital Index; IRR = Index of Relative Rurality; Demand = Average cognitive demand of reported hobbies; Number = Number of cognitive hobbies; −2 LL = −2 Log Likelihood. Adjusted for sex, age, project, adopted status, race, and ethnicity, live together, educational attainment, and occupation. Bolded = $p < .05$.

***IRR and activity-IQ association.*** Parameter estimates for the interaction of IRR with cognitive engagement on FSIQ are reported under Model 3 (see Table 3). Adding the IRR interaction with HPW or cognitive demand did not improve fit ($\Delta\chi^2$(1)<=1.7, p>=.192) for either model. Similar patterns were observed for PIQ and VIQ (S13-S16 Tables in S1 File). We did not find moderation of IRR by HPW (B=−0.61, p=.86) or cognitive demand (B=16.59, p=.18) (see Table 3).
***SCI and IRR Moderation.*** The interaction of SCI with IRR was significant after considering quantitative (B=−0.48, p=.009) and qualitative (B=−0.69, p=.008) cognitive engagement, with the main effect for SCI reaching significance (B=0.10, p=.01) in the qualitative engagement model (see Table 3, Model 4). Model fit improved with the inclusion of SCI-IRR interaction ($\Delta\chi^2$(1)>=6.8, p<=.009) and AIC was lowest of all models tested. Generally, SCI and SCI-IRR moderation associations on VIQ and PIQ resembled FSIQ, except for nonsignificant moderation on VIQ (p≥.15) (see Supplemental Materials in S1 File). Notably, implementing a Hommel multiple-testing correction [64] on the p-values for Model 4 did not alter conclusions (FSIQ and PIQ adjusted p-values=.016 to.027).

Fig 3A depicts the SCI-IRR interaction on FSIQ controlling for cognitive demand and Fig 3B plots FSIQ by cognitive demand that vary on SCI and IRR. As Fig 3A shows, rural residents differ little on FSIQ across SCI (1.04 points) compared to urban residents (7.09 points). A reversal in the cognitive advantage is seen between urban and rural residents based on SCI. Urban residents tend to score 3.86 *more* points at the upper plotted levels of SCI than more rural residing individuals. When SCI is lower, urban residents tend to score 2.19 *less* points than their rural counterparts. Fig 3B shows the FSIQ advantage of participating in more cognitive demanding activities and the FSIQ performance difference between urban residents that differ on SCI. Urban residents engaging in more demanding cognitive activities tend to score 5 more points on FSIQ after controlling for IRR and SCI. Individuals living in more urban counties with lower SCI, and rural residents irrespective of SCI, have more comparable FSIQ performance (<1.2 points).

***Sensitivity Analysis: Adjusting for Adolescent IQ.*** We repeated final analytical models to evaluate the extent to which earlier life cognitive differences explain activity or place associations on IQ performance. Best-fitting models were rerun on the subsample that included IQ16 (base model) with the addition of IQ16, controlling for years difference between adolescent and CATSLife assessments. Parameter estimates and model fit are shown in Table 4 for FSIQ and PIQ; see S17 Table in S1 File for VIQ. A general pattern emerged, IQ16 fully attenuated most activity and place associations across current IQ scores. The only measure that consistently persisted, post-adolescent IQ adjustment, was the quantitative engagement measure (i.e., HPW). Across all current IQ scores, the more time spent in leisure activities remained significantly associated with better IQ performance, but effects diminished by half ($B_{FSIQ} = 3.27$ to $B_{FSIQ} = 1.52$) after considering IQ16 scores. Place effects did not remain as robust; IQ16 fully diminished effects apart from the moderation effect of SCI and IRR on PIQ in models with quantitative activity engagement. Fig 4 illustrates the selection effects on activity and place influences on FSIQ and PIQ performance. As seen in the figure, activity influences on IQ remain, albeit effects are modest. Likewise, place influences on PIQ are also weakly related. These attenuation patterns suggest selection factors explain most underlying correlated environmental-cognitive associations and emphasize the greater importance of self-selection for the level and type of cognitive engagement.

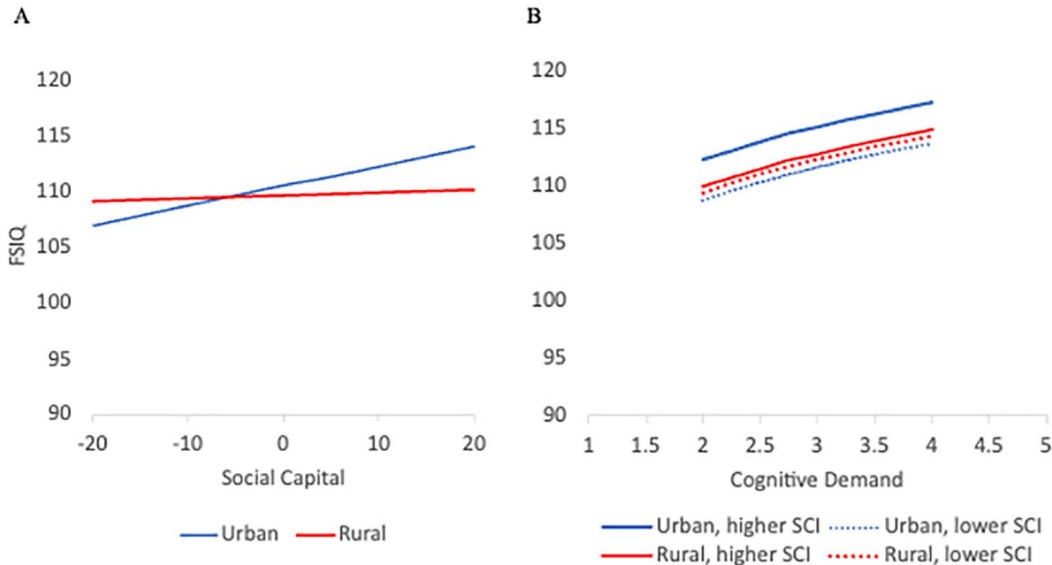

**Fig 3. Full-Scale IQ (FSIQ) associations considering cognitive demand and interaction between Social Capital (SCI) and Index of Relative Rurality (IRR). Legend: (A)** FSIQ plotted by the interaction between SCI and IRR while controlling for cognitive demand. **(B)** FSIQ by cognitive demand varying on SCI and IRR. Figures plot over 95% of the sample, Urban = 0.24 IRR score, Rural = 0.46 IRR score, Higher SCI = 10 and Lower SCI = −10.

**Table 4. Sensitivity analysis: Parameter estimates and model fit for IQ, pre- and post-adolescent IQ adjustment.**

| | FSIQ | | | | PIQ | | | |
| --- | --- | --- | --- | --- | --- | --- | --- | --- |
| | Base | | Add IQ16 | | Base | | Add IQ16 | |
| | B | se | B | se | B | se | B | se |
| **HPW Cognitive** | | | | | | | | |
| Intercept | **107.45** | 1.59 | **105.4** | 1.04 | **108.9** | 1.97 | **107.71** | 1.48 |
| HPW | **3.27** | 0.38 | **1.52** | 0.27 | **2.88** | 0.48 | **1.44** | 0.38 |
| SCI | 0.03 | 0.03 | −0.02 | 0.02 | 0.03 | 0.04 | −0.02 | 0.03 |
| IRR | −2.49 | 2.57 | 0.78 | 1.78 | −1.54 | 3.32 | 0.35 | 2.56 |
| SCI* IRR | **−0.46** | 0.18 | −0.18 | 0.13 | **−0.61** | 0.24 | **−0.50** | 0.19 |
| Year diff | | | **0.79** | 0.16 | | | **0.89** | 0.23 |
| IQ16 | | | **0.74** | 0.02 | | | **0.70** | 0.02 |
| Model Fit | | | | | | | | |
| −2 LL | 8203.4 | | 7290.4 | | 8719.9 | | 8126.7 | |
| AIC | 8247.4 | | 7336.4 | | 8761.9 | | 8174.7 | |
| Model Comparison | | | | | | | | |
| Δχ² (df) | -- | | 913.0 (2) | | -- | | 593.2 (2) | |
| p | -- | | <.0001 | | -- | | <.0001 | |
| **Cognitive Demand** | | | | | | | | |
| Intercept | **109.85** | 2.01 | **106.92** | 1.30 | **110.90** | 2.41 | **109.14** | 1.83 |
| Demand | **6.81** | 1.58 | 1.89 | 1.07 | **6.07** | 1.99 | 2.03 | 1.54 |
| Number | **3.09** | 0.86 | 0.82 | 0.61 | 1.04 | 1.11 | −0.48 | 0.87 |
| SCI | **0.09** | 0.04 | −0.03 | 0.03 | **0.10** | 0.05 | −0.05 | 0.04 |
| IRR | −4.47 | 3.23 | 0.85 | 2.20 | −2.45 | 4.09 | 1.44 | 3.14 |
| SCI* IRR | **−0.72** | 0.26 | −0.30 | 0.18 | **−0.96** | 0.33 | −0.40 | 0.25 |
| Year diff | | | **0.74** | 0.19 | | | **0.92** | 0.27 |
| IQ16 | | | **0.75** | 0.02 | | | **0.71** | 0.03 |
| Model Fit | | | | | | | | |
| −2 LL | 5173.3 | | 4582.6 | | 5477.8 | | 5098.8 | |
| AIC | 5217.3 | | 4630.6 | | 5523.8 | | 5146.8 | |
| Model Comparison | | | | | | | | |
| Δχ² (df) | -- | | 590.7 (2) | | -- | | 379.0 (2) | |
| p | -- | | <.0001 | | -- | | <.0001 | |

*Note*. FSIQ = Full-Scale IQ; PIQ = Performance IQ; HPW = Log-transformed cognitive hours per week; SCI = Social Capital Index; IRR = Index of Relative Rurality; Demand = Average cognitive demand of reported hobbies; Number = Number of cognitive hobbies; Year diff = Year difference between CATSLife and Year 16 assessments, centered at 15 years; IQ16 = Adolescent FSIQ or PIQ assessed at approximately 16 years of age, centered at 100; −2 LL = −2 Log Likelihood. Adjusted for sex, age, project, adopted status, race, and ethnicity, live together, educational attainment, and occupation. Bolded = p < .05.

## Discussion

We evaluated how community and geographic contexts may influence activity engagement-cognitive ability associations at the verge of midlife with implications to cognitive aging. In general, we observed that cognitive activity engagement had the strongest main effect influence on IQ, but meaningful differences emerged between this relationship when social capital and broader geography were considered. We found social capital associations with IQ performance were moderated by rurality, where benefits of higher social capital on IQ were maximized for those living in more urban than more rural areas. Moreover, selection effects were supported via attenuation of rurality associations due to individual level SES indices and adolescent IQ, and via patterns of sibling similarity for cognitive engagement, social capital, and rurality. Hence,

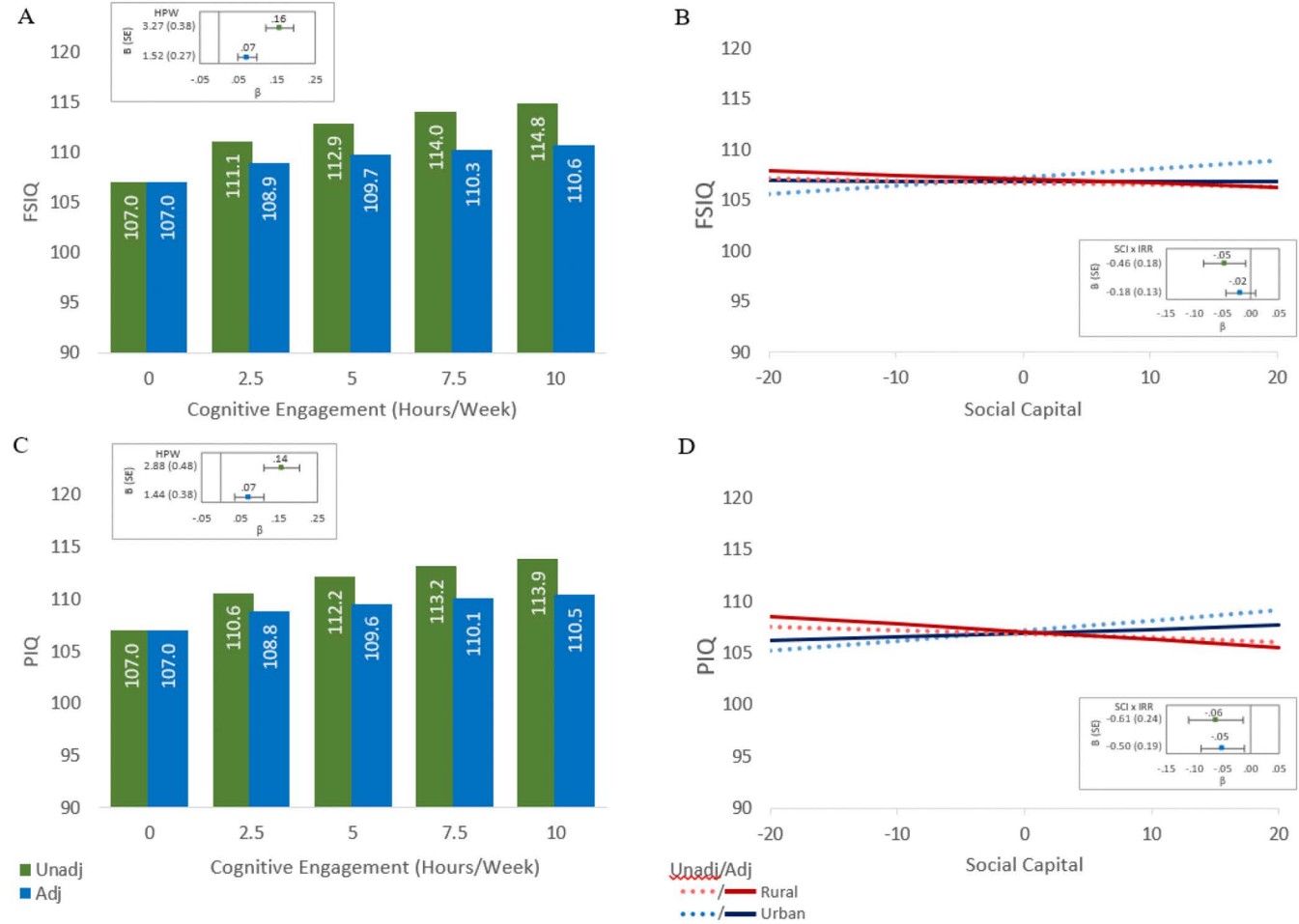

**Fig 4. Full-scale and Performance IQ scores by Cognitive Engagement and Place, Pre- and post-adolescent IQ adjustment. Legend:** FSIQ = Full-scale IQ, PIQ = Performance IQ, SCI x IRR = Social capital (SCI) and rurality (IRR) interaction on IQ, Unadj/Adj = Unadjusted/Adjusted for adolescent IQ (IQ16). Panels A and C depict FSIQ and PIQ performance by cognitive engagement before (green) and after (blue) adjusting for IQ16. Panels B & D depict FSIQ and PIQ performance by social capital and rurality before (dashed) and after (solid) IQ16 adjustment. More rural (IRR = 0.46) is shown in red and more urban (IRR = 0.24) in blue. Inset panel forest plots show standardized estimates with 95% confidence intervals for cognitive engagement (Panels A & C) and SCI x IRR effects on FSIQ or PIQ (Panels B & D) before (green) and after (blue) adjusting for IQ16; see Table 4 for unstandardized estimates.

our cross-sectional findings suggest dynamic and co-acting systems underlie lifestyle and environmental factors related to individual differences in cognitive health and maintenance at midlife. Longitudinal follow-up will be necessary to evaluate how individual lifestyle factors and place effects may codevelop to alter cognitive maintenance and health.

The engagement-cognitive associations we found adds to the growing literature suggesting benefits of cognitive engagement to cognitive functioning across adulthood [2,14]. The interrelation between engagement and IQ remained robust after considering individual education and occupational prestige, suggesting that the individual differences in education or occupational attainment do not explain the links we observed. Research has suggested that engagement in intellectually stimulating activities during mid-life may moderate the relationship between mid-life cognitive ability and functional brain network segregation, independent of education and occupation [65]. Additionally, the inferred genetic influences contributing to engagement when comparing the sibling intraclass correlations aligns with heritable influences

reported for intellectual leisure activities [66]. The genetic influences underlying engagement may indicate that individuals engage in activity selection that aligns in part with genetic propensity. This would be supportive of our findings that earlier life cognitive differences had a greater influence on diminishing the subjective rather than the objective measure of leisure activity engagement influence on IQ; or that individual preference is better exemplified by the level of engagement in cognitively demanding activities than the general time an individual reports spending. Moreover, interpretations of activity selection are consistent with findings that cognitive engagement is accounted for by early life factors and abilities [2,25,26].

We found that urban residents tend to have a cognitive advantage compared to more rural residents, which was expected [3–5,67,68]. A key finding from our work is that rurality moderated social capital associations with IQ performance. Cognitive differences between rural and urban residents have been speculated to represent educational attainment gaps across the locales [5] but perhaps, some of the inconsistent cognitive findings are attributable to studies not considering the intersection of social capital with rurality. As our work suggests, urban residents were more sensitive to the influences of social capital on IQ than their rural counterparts which IQ performance remained relatively stable across social capital. For urban residents, IQ was predicted to differ by nearly a half a standard deviation between locales low (IQ = 106.9) or high (IQ = 114.0) on social capital (see Fig 3a), after controlling for cognitive demand. Moreover, social capital and rurality did not alter activity-IQ associations. Given a paucity of literature to inform this series of analyses, we propose a speculative interpretation. It could be that the population size and density of urban places may require greater local and civic intervention to build community and increase social connectedness between residents. For example, urban areas are likely to have greater availability of civic, social, and business associations, as well as political and professional organizations, all of which are associated with higher social capital [17]. Urban areas with greater social capital may also provide more opportunities to engage in activities that could benefit cognitive reserve [11,69]. However, when these organizations, events, or programs are lacking for urban residents as illustrated when social capital is low, individuals may feel less attached, have fewer opportunities for local engagement, or participate in less activities, and thus adversely influence cognitive performance. This may explain why the cognitive performance differences between urban residents seem to be more sensitive to social capital than rural residents where independent or limited group engagement such as reading or playing cards are more likely [53,56,70–72]. While we found the moderating influence of social capital and rurality on IQ performance, these associations diminished for full-scale after accounting for adolescent IQ suggesting underlying selection factors contributing to these connections. Performance IQ however retained significance for social capital and rurality which is consistent with other work in CATSLife and other samples showing the salience of perceived neighborhood stressors on spatial performance indexed by Block Design a WAIS-III subtest [73] as well as for a processing speed task [74].

Factors influencing individuals' choice of environments like SES, which is interconnected with earlier life [75] and later life cognitive differences [9], may further shape the interplay of place effects we found. Individuals with higher levels of educational attainment are more likely to reside in urban areas due to better opportunities for employment [68,76], and education is the strongest correlate with social trust and membership in a broader range of groups [77]. Moreover, education has large positive associations with the creation of trust among members of the community and engagement in civic activities [21,78], such as voting, volunteering or even community gardening. Further, higher educational attainment may be a delaying factor for dementia onset beyond genetic factors [79] with a possible pathway via greater engagement in cognitively stimulating activities across the lifespan [80,79]. It is possible that the combination of higher education, greater participation in cognitively stimulating activities and living in an urban environment increases the social capital in the area, which points to a possible environmental selection [81]. On the other hand, areas with higher social capital provide more opportunities to achieve higher education [17] and engage in more cognitively stimulating activities. These prior results as well as the aforementioned work on geospatial selection noted above suggest a complex picture of selection that should be considered in future work that includes individual and geospatial information.

Indeed, rurality-cognitive associations were attenuated by over 67% after individual level SES adjustment, supportive of environmental selection. A longitudinal study of Add Health participants (ages 24–32), that overlaps with CATSLife ages, observed that those with higher educational attainment polygenic scores tended to sort into neighborhoods with higher population densities and with higher average educational attainment [81]. While the rural-urban educational attainment gap may be narrowing in recent cohorts altering dementia risk [5], ongoing disadvantages may still be apparent including under-diagnosis or later-stage diagnosis of Alzheimer's disease and related dementias [82]. Moreover, early life factors may accumulate to affect cognitive performance in later life with residence being an important consideration [3,67,80]. Longitudinal analysis of community and geographic influences would inform whether and how early and mid-life environments affect cognitive performance proximally and distally.

Selection factors may play out in complex ways across contexts. In contrast to cognitive engagement with suggestions of gene-environment associations, we found shared environmental contributions to county-level social capital and urbanicity-rurality based on intraclass correlation patterns. Our SCI findings are consistent with findings for neighborhood deprivation, a structural community measure [83], in contrast to individual perceptions of social trust, which shows greater genetic variance (66%) for adolescents and young adults [84], consistent with patterns of personality characteristics [85]. We observed greater SCI similarity among twin pairs than non-twin siblings possibly indicating a special twin environment whereby twins may choose to live closer to each other [86] or simply due to age cohort differences; CATSLife twins were born nearly a decade after the control/adopted siblings [41,42]. Even though few CATSLife twins live together, housing market changes over the last 20 years [87] may influence residential choice and hence distance between twins versus non-twin siblings. Like SCI, we found shared environmental contributions in urbanicity-rurality, consistent with a study of Australian twins which showed shared environmental influences were the largest contributor to choice of residence ranging from urban, suburban, to nonurban with diminishing influence for older aged cohorts [88]. These findings of strong but waning influences of shared environmental contributions in urbanicity across age were replicated directly in a Dutch twin sample [86] and indirectly with another environmental metric, walkability, before and after controlling for social deprivation [89].

Our study helps advance knowledge about the interrelationship between individual and contextual influences on cognition, yet our study is not without limitations. The use of cross-sectional data and self-reported measures (e.g., cognitive engagement) limits our ability to infer causality and introduces potential biases, such as recall inaccuracies and socially desirable responding, which may obscure true associations with cognitive functioning. Our sample is more educated and has higher average IQ scores than the general population; nonetheless, there is reasonable variability in educational attainment and IQ distribution collectively. Even though we evaluated multiple measures of mentally stimulating activities (i.e., time, demand, and number), we could not examine the subjective experience related to leisure activity engagement nor did we explore the dynamics between other domains of engagement [2]. Future work is warranted that further explores recreational engagement accounting for the individual approach and demand of their leisure activities and hobbies. Although missing data was minimal and did not substantially impact the analyses, it is important to note that a small number of participants did not report codable hobbies (N = 130), and an even smaller group lacked occupational attainment data (N = 50), which may slightly limit generalizability for those variables. While more nuanced and individualized data can be gained from qualitative responses it may come with a greater risk of missingness or incompleteness, that may be associated with sociodemographic factors. Even so, we have generally consistent results between the cognitive engagement measures, indicating comparable findings. Additionally, we could not account for both family and geospatial clustering due to the large geographic spread outside of Colorado. Last, our sample is less diverse than the US as a whole (92% versus 62% white) [90], hence we could not examine if association patterns of social capital access and rurality vary by race or ethnicity.

In conclusion, our study explored the interrelationship between individual engagement, community, and geography on cognitive performance. Our primary findings are three-fold that raise further implications and questions. First, engagement

in cognitive leisure activities, both in amount of time spent and the cognitive demand of hobbies, was significantly associated with cognitive performance which did not vary by geography. Secondly, small associations of rurality and cognitive performance were observed but attenuated when accounting for sibling structure, educational, and occupational attainment, indicative of environmental selection [80]. Third, social capital and rurality were not independently associated with IQ performance, after accounting for covariates and cognitive engagement. Rather we found social capital and cognitive associations were moderated by rurality, suggestive of important interplay between community and geography. Urban dwelling residents' cognitive functioning was more sensitive to variation in social capital than their rural counterparts, where the benefits of living in an urban environment were only magnified when social capital was high. These locale-based patterns may underline person and place effects seen in cognitive performance and future research should consider individual behavior, community, and geography to better elucidate the nuance and diverse patterns in cognitive health. Our work illustrates that individual factors and socio-geographic environments matter and suggests that a fuller appreciation of person-environment dynamics is needed to understand geographic differences in cognitive performance.

## Supporting information

**S1 File. PLOSOne Supplement Description and Tables.** Supplemental Materials for Cognitive Functioning in Context: Leisure Activity Engagement, Social Capital, and Urbanicity-Rurality Interplay Manuscript.
(DOCX)

**S1 Fig. Verbal IQ (VIQ) associations considering cognitive demand and interaction between Social Capital (SCI) and Index of Relative Rurality (IRR).** *Note.* (A) VIQ plotted by the interaction between SCI and IRR while controlling for cognitive demand. (B) VIQ by cognitive demand varying by SCI and IRR levels. Urban is an IRR score of 0.24, Rural is an IRR score of 0.46, Higher SCI = 10 and lower SCI = −10.
(TIF)

**S2 Fig. Performance IQ (PIQ) associations considering cognitive demand and interaction between Social Capital (SCI) and Index of Relative Rurality (IRR).** *Note.* (A) PIQ plotted by the interaction between SCI and IRR while controlling for cognitive demand. (B) PIQ by cognitive demand varying by SCI and IRR levels. Urban is an IRR score of 0.24, Rural is an IRR score of 0.46, Higher SCI = 10 and lower SCI = −10.
(TIF)

## Acknowledgments

We thank the CATSLife staff and participants who have graciously participated over many years.

## Author contributions

**Conceptualization:** Paige Trubenstein.

**Data curation:** Shandell Pahlen, Robin P Corley, Chandra A Reynolds.

**Formal analysis:** Paige Trubenstein, Shandell Pahlen.

**Funding acquisition:** Sally J Wadsworth, Chandra A Reynolds.

**Investigation:** Paige Trubenstein, Shandell Pahlen, Robin P Corley, Chandra A Reynolds.

**Methodology:** Paige Trubenstein, Chandra A Reynolds.

**Project administration:** Robin P Corley, Sally J Wadsworth, Chandra A Reynolds.

**Software:** Shandell Pahlen.

**Supervision:** Sergio Rey, Sally J Wadsworth, Chandra A Reynolds.

**Visualization:** Paige Trubenstein, Shandell Pahlen.

**Writing – original draft:** Paige Trubenstein, Shandell Pahlen.

**Writing – review & editing:** Paige Trubenstein, Shandell Pahlen, Robin P Corley, Sergio Rey, Sally J Wadsworth, Chandra A Reynolds.

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
