## [Decision Letter · Decision Letter 0]

1 Oct 2025

Dear Dr. Trubenstein,

Thank you for submitting your manuscript to PLOS ONE. After careful consideration, we feel that it has merit but does not fully meet PLOS ONE’s publication criteria as it currently stands. Therefore, we invite you to submit a revised version of the manuscript that addresses the points raised during the review process.

**For your revised version, please address the comments below from the Reviewer and Academic Editor.**

We look forward to receiving your revised manuscript.

Kind regards,

Elise Rivera

Academic Editor

PLOS ONE

**Journal Requirements:**

1. When submitting your revision, we need you to address these additional requirements. Please ensure that your manuscript meets PLOS ONE's style requirements, including those for file naming. The PLOS ONE style templates can be found at https://journals.plos.org/plosone/s/file?id=wjVg/PLOSOne_formatting_sample_main_body.pdf and https://journals.plos.org/plosone/s/file?id=ba62/PLOSOne_formatting_sample_title_authors_affiliations.pdf 2. Thank you for stating the following financial disclosure: National Institutes of Health, NIH AG046938 [MPIs, Reynolds (Contact), Wadsworth]   Please state what role the funders took in the study.  If the funders had no role, please state: "The funders had no role in study design, data collection and analysis, decision to publish, or preparation of the manuscript." If this statement is not correct you must amend it as needed. Please include this amended Role of Funder statement in your cover letter; we will change the online submission form on your behalf. 3. Thank you for stating the following in the Acknowledgments Section of your manuscript: The authors gratefully acknowledge support from the National Institutes of Health, NIH AG046938 [MPIs, Reynolds (Contact), Wadsworth]. The content is solely the responsibility of the authors and does not necessarily represent the official views of the National Institutes of Health. We thank the CATSLife staff and participants who have graciously participated over many years. We note that you have provided funding information that is not currently declared in your Funding Statement. However, funding information should not appear in the Acknowledgments section or other areas of your manuscript. We will only publish funding information present in the Funding Statement section of the online submission form. Please remove any funding-related text from the manuscript and let us know how you would like to update your Funding Statement. Currently, your Funding Statement reads as follows: National Institutes of Health, NIH AG046938 [MPIs, Reynolds (Contact), Wadsworth]  Please include your amended statements within your cover letter; we will change the online submission form on your behalf. 4. Thank you for stating the following in your Competing Interests section:  “NO authors have competing interests”  Please complete your Competing Interests on the online submission form to state any Competing Interests. If you have no competing interests, please state "The authors have declared that no competing interests exist.", as detailed online in our guide for authors at http://journals.plos.org/plosone/s/submit-now This information should be included in your cover letter; we will change the online submission form on your behalf. 5. When completing the data availability statement of the submission form, you indicated that you will make your data available on acceptance. We strongly recommend all authors decide on a data sharing plan before acceptance, as the process can be lengthy and hold up publication timelines. Please note that, though access restrictions are acceptable now, your entire data will need to be made freely accessible if your manuscript is accepted for publication. This policy applies to all data except where public deposition would breach compliance with the protocol approved by your research ethics board. If you are unable to adhere to our open data policy, please kindly revise your statement to explain your reasoning and we will seek the editor's input on an exemption. Please be assured that, once you have provided your new statement, the assessment of your exemption will not hold up the peer review process. 6. Please amend either the title on the online submission form (via Edit Submission) or the title in the manuscript so that they are identical. 7. Your ethics statement should only appear in the Methods section of your manuscript. If your ethics statement is written in any section besides the Methods, please move it to the Methods section and delete it from any other section. Please ensure that your ethics statement is included in your manuscript, as the ethics statement entered into the online submission form will not be published alongside your manuscript. 8. Please include captions for your Supporting Information files at the end of your manuscript, and update any in-text citations to match accordingly. Please see our Supporting Information guidelines for more information: http://journals.plos.org/plosone/s/supporting-information. 9. If the reviewer comments include a recommendation to cite specific previously published works, please review and evaluate these publications to determine whether they are relevant and should be cited. There is no requirement to cite these works unless the editor has indicated otherwise. 

**Additional Editor Comments:**

This is a well written manuscript that addresses and under-researched topic. These following are comments to address to enhance the quality of this manuscript and suitability for publication.

Introduction: The intro is very well written. While there is some mention of applying theory, I think the intro could be enhanced by further discussing the theoretical underpinnings of this research (e.g., bioecological model, socio-ecological model etc). Just expanding on this to make it a bit more explicit to the readers would suffice.

Measure (Page 7): Could the authors comment on the psychometric properties of the selected measures for cognitive ability, cognitive engagement, community and environmental measures, educational attainment, and occupational attainment.

Methods: Given the large number of models run in the analyses, were there adjustments made for multiple testing given that a high number of models increases the risk of type 1 error. It is advised to include an explicit statement about this and support why or why not this was adjusted for. These might be useful articles for this:

Bender R, Lange S. Adjusting for multiple testing—when and how? Journal of Clinical Epidemiology. 2001;54(4):343-9.

Feise RJ. Do multiple outcome measures require p-value adjustment? BMC Medical Research Methodology. 2002;2(8):1-4.

Discussion: I would add a limitation about the cross-sectional nature of the study design and self report measures.

Reviewers' comments:

Reviewer's Responses to Questions

**Comments to the Author**

1. Is the manuscript technically sound, and do the data support the conclusions?

Reviewer #1: Yes

2. Has the statistical analysis been performed appropriately and rigorously?

Reviewer #1: I Don't Know

3. Have the authors made all data underlying the findings in their manuscript fully available?

Reviewer #1: Yes

4. Is the manuscript presented in an intelligible fashion and written in standard English?

Reviewer #1: Yes

**Reviewer #1:** The manuscript investigates how the relationship between leisure activities and cognitive function in midlife may be influenced by contextual factors such as residential setting (rural vs. urban) and community-related resources or social support. The study is well structured and addresses an important and timely research question. A particular strength is the focus on midlife participants, rather than exclusively older adults, as this provides valuable insights into early intervention opportunities for promoting healthy cognitive ageing. Another notable strength is the multidimensional assessment of leisure activity participation. Rather than limiting the evaluation to activity frequency, the authors have also considered cognitive demand and the duration of engagement, which adds depth to the findings. This approach is methodologically stronger than frequency-only measures, which can miss important aspects of activity quality. Overall, the manuscript makes a meaningful contribution to the field by extending knowledge on how contextual and activity-related factors intersect in shaping cognitive outcomes.

I believe that the manuscript could be further improved by considering the following major and minor suggestions. These recommendations are intended to strengthen the clarity, rigour, and overall contribution of the study.

Major Issues

1) Adherence to reporting guidelines (STROBE)

While the supplementary materials provide comprehensive supporting data, the manuscript would benefit from including a completed STROBE checklist as a supplementary file. This will demonstrate adherence to recognised reporting standards for observational studies and improve transparency for editors and readers.

Missing data acknowledgment

2) Although the amount of missing data is relatively small and does not substantially affect the analyses, it is recommended that the authors acknowledge this as a limitation in the Discussion section. Providing information on which variables had missing data and their proportion would enhance transparency and contextualise the findings.

Minor Issues

1) Cognitive reserve in the Introduction

The Introduction briefly mentions cognitive reserve, but the manuscript could more clearly highlight that the study assesses key components of cognitive reserve (IQ, educational attainment, and occupational complexity) as measured in the Methods section. This would help contextualize the study’s rationale and can also be revisited in the Discussion when interpreting the findings.

2)Clarity of the Results section for non-specialist readers

To improve readability, especially for readers outside the field, it would be helpful to explicitly indicate which analyses correspond to each of the four research questions. Additionally, clearly stating what each finding indicates (e.g., “this result suggests…”) would make the interpretation of results more transparent and accessible.

**Do you want your identity to be public for this peer review?** For information about this choice, including consent withdrawal, please see our Privacy Policy

Reviewer #1: No

While revising your submission, please upload your figure files to the Preflight Analysis and Conversion Engine (PACE) digital diagnostic tool, https://pacev2.apexcovantage.com/ . PACE helps ensure that figures meet PLOS requirements. To use PACE, you must first register as a user. Registration is free. Then, login and navigate to the UPLOAD tab, where you will find detailed instructions on how to use the tool. If you encounter any issues or have any questions when using PACE, please email PLOS atfigures@plos.org

---

## [Author Response · Author response to Decision Letter 1]

5 Dec 2025

a. We now adhere to the PLOS ONE's style requirements for file naming and elsewhere.

2. Thank you for stating the following financial disclosure: National Institutes of Health, NIH AG046938 [MPIs, Reynolds (Contact), Wadsworth]. Please state what role the funders took in the study.

a. We have now added the following sentence regarding the funders role under the financial disclosure section online: “The content is solely the responsibility of the authors and does not necessarily represent the official views of the National Institutes of Health.”

b. Thus, our full financial disclosure statement now reads: “The authors gratefully acknowledge support from the National Institutes of Health, NIH AG046938 [MPIs, Reynolds (Contact), Wadsworth]. The content is solely the responsibility of the authors and does not necessarily represent the official views of the National Institutes of Health.”

3. Please remove any funding-related text from the manuscript and let us know how you would like to update your Funding Statement.

a. Funding information from the manuscript has been removed and our full statement, including amendments, is provided above.

4. Please complete your Competing Interests on the online submission form to state any Competing Interests. If you have no competing interests, please state "The authors have declared that no competing interests exist."

a. We now so note "The authors have declared that no competing interests exist” on the online submission form.

5. When completing the data availability statement of the submission form, you indicated that you will make your data available on acceptance.

a. Yes, see the included Data Availability Statement in the Manuscript (page 27).

6. Please amend either the title on the online submission form (via Edit Submission) or the title in the manuscript so that they are identical.

a. The titles have been edited to be identical.

7. Your ethics statement should only appear in the Methods section of your manuscript.

a. The ethics statement has been moved to be within the methods section of the manuscript (page 8).

8. Please include captions for your Supporting Information files at the end of your manuscript, and update any in-text citations to match accordingly.

a. Figure Captions have now been added to the end of the manuscript (see Pages 38-39), with matching in-text citations.

9. If the reviewer comments include a recommendation to cite specific previously published works, please review and evaluate these publications to determine whether they are relevant and should be cited.

a. No specific citation recommendations were made by reviewers.

a. We have reviewed our reference list and ensured that all citations made in text appear in the reference list and that all in the list are cited in the text. We have not cited any retracted citations as of the submission of this revised manuscript.

This is a well written manuscript that addresses and under-researched topic. These following are comments to address to enhance the quality of this manuscript and suitability for publication.

We thank you for the positive feedback and we believe that following your suggestions has improved the paper as noted below in our specific responses.

1. Introduction: The intro is very well written. While there is some mention of applying theory, I think the intro could be enhanced by further discussing the theoretical underpinnings of this research (e.g., bioecological model, socio-ecological model etc). Just expanding on this to make it a bit more explicit to the readers would suffice.

a. We thank you for the positive feedback, we have now added an expansion of the bioecological model to expand on the underlying theory (see page 3).

2. Measure (Page 7): Could the authors comment on the psychometric properties of the selected measures for cognitive ability, cognitive engagement, community and environmental measures, educational attainment, and occupational attainment.

a. We have included additional background information on the construction and development of several of the measures we used, including IQ (page 7), cognitive engagement (pages 8-9), SCI (pages 9-10), and IRR (page 10). In addition, we added a table of partial correlations among education, occupational complexity, cognitive engagement, social capital and rurality variables (Table S4, referred to on page 15 in text): “Partial correlations among education, occupational complexity, cognitive engagement, social capital and rurality variables are reported in Table S4.”

3. Methods: Given the large number of models run in the analyses, were there adjustments made for multiple testing given that a high number of models increases the risk of type 1 error. It is advised to include an explicit statement about this and support why or why not this was adjusted for. These might be useful articles for this:

Bender R, Lange S. Adjusting for multiple testing—when and how? Journal of Clinical Epidemiology. 2001;54(4):343-9.

Feise RJ. Do multiple outcome measures require p-value adjustment? BMC Medical Research Methodology. 2002;2(8):1-4.

a. Thank you for this suggestion. In many of our models fitted, we check for robustness of associations when particular selection factors or covariates are added. These are not the kinds of tests where we would employ a multiple testing correction. However, our tests of moderation by IRR warrants examination. We observed significance for IRR*SCI in models evaluating cognitive engagement (HPW or Demands) and thus for these tests we implemented the Hommel adjustment (Hommel, 1988) on FSIQ, VIQ, and PIQ outcomes (6 tests). No differences in conclusions are apparent (see Table below).

FSIQ VIQ PIQ

Model 4 (1 df) HPW Demands HPW Demands HPW Demands

Unadjusted p-values 0.009 0.009 0.206 0.147 0.007 0.004

Hommel p-value 0.027 0.026 0.206 0.206 0.020 0.016

b. We added a note on pages 17-18 that we achieve no difference in conclusions with such an adjustment: “Notably, implementing a Hommel multiple-testing correction on the p-values for Model 4 did not alter conclusions (FSIQ and PIQ adjusted p-values = .016 to .027).”

The manuscript investigates how the relationship between leisure activities and cognitive function in midlife may be influenced by contextual factors such as residential setting (rural vs. urban) and community-related resources or social support. The study is well structured and addresses an important and timely research question. A particular strength is the focus on midlife participants, rather than exclusively older adults, as this provides valuable insights into early intervention opportunities for promoting healthy cognitive ageing. Another notable strength is the multidimensional assessment of leisure activity participation. Rather than limiting the evaluation to activity frequency, the authors have also considered cognitive demand and the duration of engagement, which adds depth to the findings. This approach is methodologically stronger than frequency-only measures, which can miss important aspects of activity quality. Overall, the manuscript makes a meaningful contribution to the field by extending knowledge on how contextual and activity-related factors intersect in shaping cognitive outcomes. I believe that the manuscript could be further improved by considering the following major and minor suggestions. These recommendations are intended to strengthen the clarity, rigor, and overall contribution of the study.

We thank you for the very positive feedback and constructive points and we detail our specific responses to your helpful suggestions.

Major Issues

1) Adherence to reporting guidelines (STROBE)

While the supplementary materials provide comprehensive supporting data, the manuscript would benefit from including a completed STROBE checklist as a supplementary file. This will demonstrate adherence to recognised reporting standards for observational studies and improve transparency for editors and readers.

a. A STROBE checklist was completed and submitted as a supplementary file.

Missing data acknowledgment

2) Although the amount of missing data is relatively small and does not substantially affect the analyses, it is recommended that the authors acknowledge this as a limitation in the Discussion section. Providing information on which variables had missing data and their proportion would enhance transparency and contextualise the findings.

a. We now note missing data as a limitation in the Discussion section (see page 25). Moreover, we note which variables had missing data and their proportion (pages 7-8, 16-17, & 25, Table 1, Table S2). Missingness between IQ and HPW is very minimal. Only missingness for those that didn’t report clearly codable hobbies may impact the follow-up analysis (N=130; 15% of those reporting spending any time on a hobby). In other words, people who engage in hobby time, but their qualitative responses weren’t codable for a variety reasons (skipped, unidentifiable abbreviations, non-hobbies/joke response) were excluded from analyses and also tend to have lower educational attainment and perform less well on IQ battery tasks (see page 8). It remains unclear how their inclusion could impact results, but this work does highlight the unique challenges in the level of generalizability involved in qualitative compared to quantitative self-reported data. Thus, while more nuanced and individualized data can be gained from qualitative responses it may come with a greater risk of missingness or incompleteness, that may be associated with sociodemographic factors. Even so, we have generally consistent results between the measures, indicating a similar story.

Minor Issues

1) Cognitive reserve in the Introduction: The Introduction briefly mentions cognitive reserve, but the manuscript could more clearly highlight that the study assesses key components of cognitive reserve (IQ, educational attainment, and occupational complexity) as measured in the Methods section. This would help contextualize the study’s rationale and can also be revisited in the Discussion when interpreting the findings.

a. We now clearly highlight that we address key components of cognitive reserve in the introduction by adding: “Cognitive reserve is conceptualized as the brain’s capacity to flexibly recruit and optimize cognitive processes, accounting for individual differences in vulnerability to cognitive impairment and is often measured through proxies such as IQ, educational attainment, occupational complexity, and even leisure and physical activity” (see page 3).

b. To the discussion we added: “Urban areas with greater social capital may also provide more opportunities to engage in activities that could benefit cognitive reserve” (see page 22).

2) Clarity of the Results section for non-specialist readers. To improve readability, especially for readers outside the field, it would be helpful to explicitly indicate which analyses correspond to each of the four research questions. Additionally, clearly stating what each finding indicates (e.g., “this result suggests…”) would make the interpretation of results more transparent and accessible.

a. Thank you for these suggestions. We now explicitly indicate which analyses correspond to each of the four research questions and have added summary interpretations. In addition, for improved flow and according to journal instructions, we include the figure captions text at the end of the main document, in addition to notations in the text where the figures should be placed.

---

## [Editor Report · Decision Letter 1]

8 Dec 2025

Cognitive Functioning in Context: Leisure Activity Engagement, Social Capital, and Urbanicity-Rurality Interplay

PONE-D-25-33875R1

Dear Dr. Trubenstein,

We’re pleased to inform you that your manuscript has been judged scientifically suitable for publication and will be formally accepted for publication once it meets all outstanding technical requirements.

Kind regards,

Elise Rivera

Academic Editor

PLOS One
---

## [Editor Report · Acceptance letter]

PONE-D-25-33875R1

PLOS One

Dear Dr. Trubenstein,

I'm pleased to inform you that your manuscript has been deemed suitable for publication in PLOS One. Congratulations! Your manuscript is now being handed over to our production team.

Kind regards,

on behalf of

Dr. Elise Rivera

Academic Editor

PLOS One